# The Antidiabetic Agent Acarbose Improves Anti-PD-1 and Rapamycin Efficacy in Preclinical Renal Cancer

**DOI:** 10.3390/cancers12102872

**Published:** 2020-10-06

**Authors:** Rachael M. Orlandella, William J. Turbitt, Justin T. Gibson, Shannon K. Boi, Peng Li, Daniel L. Smith, Lyse A. Norian

**Affiliations:** 1Graduate Biomedical Sciences, University of Alabama at Birmingham (UAB), Birmingham, AL 35233, USA; rorlande@uab.edu (R.M.O.); jtgibson@uab.edu (J.T.G.); Shannon.Boi@STJUDE.ORG (S.K.B.); 2Department of Nutrition Sciences, UAB, Birmingham, AL 35233, USA; wturbitt@uab.edu (W.J.T.); dsmithjr@uab.edu (D.L.S.J.); 3School of Nursing, UAB, Birmingham, AL 35233, USA; pli@uab.edu; 4Nutrition Obesity Research Center, UAB, Birmingham, AL 35233, USA; 5O’Neal Comprehensive Cancer Center, UAB, Birmingham, AL 35233, USA

**Keywords:** kidney cancer, glucose, immunity, nutrition, immunotherapy

## Abstract

**Simple Summary:**

Although immune-stimulatory and targeted therapies benefit many patients with metastatic kidney cancer, a sizeable proportion of patients fail to respond. Recent studies in mice demonstrate that nutrient-limiting dietary interventions can improve responses to chemotherapy. However, these studies did not investigate effects on metastasis, and the impact of these interventions on the response to immunotherapy or targeted therapies in kidney cancer is unknown. We therefore studied the effects of a glucose-limiting drug called acarbose, which is used to treat type 2 diabetes, in a spontaneously-metastasizing mouse model of kidney cancer. We found that acarbose slowed kidney cancer growth and promoted protective immune responses. In combination with either an immunotherapy or a targeted therapy used clinically to treat kidney cancer, acarbose led to improved treatment outcomes and reduced lung metastases. Our findings contribute to the emerging idea of using nutrition-based interventions to enhance responses to cancer treatments.

**Abstract:**

Although immune checkpoint inhibitors and targeted therapeutics have changed the landscape of treatment for renal cell carcinoma (RCC), most patients do not experience significant clinical benefits. Emerging preclinical studies report that nutrition-based interventions and glucose-regulating agents can improve therapeutic efficacy. However, the impact of such agents on therapeutic efficacy in metastatic kidney cancer remains unclear. Here, we examined acarbose, an alpha-glucosidase inhibitor and antidiabetic agent, in a preclinical model of metastatic kidney cancer. We found that acarbose blunted postprandial blood glucose elevations in lean, nondiabetic mice and impeded the growth of orthotopic renal tumors, an outcome that was reversed by exogenous glucose administration. Delayed renal tumor outgrowth in mice on acarbose occurred in a CD8 T cell-dependent manner. Tumors from these mice exhibited increased frequencies of CD8 T cells that retained production of IFNγ, TNFα, perforin, and granzyme B. Combining acarbose with either anti-PD-1 or the mammalian target of rapamycin inhibitor, rapamycin, significantly reduced lung metastases relative to control mice on the same therapies. Our findings in mice suggest that combining acarbose with current RCC therapeutics may improve outcomes, warranting further study to determine whether acarbose can achieve similar responses in advanced RCC patients in a safe and likely cost-effective manner.

## 1. Introduction

Despite advances in the use of immunotherapy and targeted therapeutics for the treatment of metastatic renal cell carcinoma (RCC), response rates remain suboptimal. Although immune checkpoint inhibitors (ICIs) against programmed cell death-1 (anti-PD-1) alone or in combination with cytotoxic T lymphocyte-associated protein 4 (CTLA-4) generate durable responses in a subset of patients, a substantial proportion (>50%) do not respond [1]. Moreover, alternative second- or third-line targeted therapies such as rapamycin-derived mTOR inhibitors (everolimus and temsirolimus) only achieve responses in <10% of patients [2]. Several factors mediate resistance to ICIs, including insufficient endogenous anti-tumor immunity and intratumoral immunosuppression, leading to immune evasion [3]. Recent findings also demonstrate that modulations in T cell anti-tumor immunity contribute to mTOR inhibitor efficacy [4], and reactivation of anti-tumor immunity promotes sustained responses in several other nonimmune-based treatments, including chemotherapy [5,6].

Additional barriers to therapeutic response include the metabolic landscape of the tumor microenvironment itself, which often features aberrant Warburg-like glycolysis of tumor cells and subsequent lactic acid buildup within tumors [7]. This harsh environment can independently impair CD8 T cell responses [8,9,10], and accumulating evidence suggests that strategies aimed at restricting glycolytic metabolism within tumors may improve anti-tumor immunity and ICI response [11,12], although other studies demonstrate impaired CD8 T cell function with glucose deprivation in tumors [7,13]. RCC in particular exhibits a distinct metabolic landscape with mutations in several metabolic genes [14] and a higher glycolytic signature compared to tumors in other anatomic sites [15]. Interestingly, elevated expression of genes indicative of glycolysis (including GLUT1 and LDHA) in human RCC tumor tissues correlate with low CD8 tumor-infiltrating lymphocyte (TIL) abundance [16]. Overall, these findings highlight the potential benefits of targeting glucose in RCC tumors to improve CD8 T cell anti-tumor immunity and ultimately, therapeutic outcomes.

Glucoregulatory agents in use for treating type II diabetes offer a well-tolerated, inexpensive, clinically relevant method to target dysregulated glucose metabolism. Metformin, perhaps the most well-known antidiabetic agent, exerts anticancer effects in a CD8 T cell-dependent manner [17] and improves anti-PD-1 efficacy [18,19]. However, the antineoplastic properties of metformin likely result from its ability to activate AMP-activated protein kinase (AMPK) [20], inhibit mitochondrial complex I [21], stimulate anti-tumor immunity, and/or reduce hypoxia within the tumor microenvironment [19] rather than its actions on glucose regulation alone.

Another glucoregulatory agent is acarbose, an orally administered competitive alpha-amylase and alpha-glucosidase inhibitor. Acarbose delays the breakdown of complex dietary carbohydrates in the brush border of the small intestine and lessens the rapid elevation in blood glucose following a meal (postprandial) [22]. In addition to its role as an FDA-approved treatment for type II diabetes and hyperglycemia, acarbose has also been investigated in longevity/healthy aging studies as a candidate caloric restriction mimetic (CRM) [23]. CRMs are agents that recapitulate the benefits of caloric restriction (i.e., longevity and delayed onset of age-related diseases) without limiting calorie intake. In support of this, acarbose extends longevity in mice [24] and is associated with reduced colorectal cancer incidence in patients with type II diabetes in a dose-dependent manner [25]. Importantly, recent reports in nonmetastatic subcutaneous murine tumor models found that CRMs improved responses to chemotherapy [5] or a combination of chemotherapy and ICIs [26] in a CD8 T cell-dependent manner. However, the effects of acarbose on tumor progression, anti-tumor immunity, and therapeutic efficacy remain unknown.

We addressed this question in a preclinical model of orthotopic kidney cancer. We show here that acarbose inhibited renal tumor growth in a CD8 T cell-dependent manner, as CD8 T cell depletion abrogated this effect. Combining acarbose with either anti-PD-1 or rapamycin improved therapeutic efficacy in mice. These findings show that combining acarbose with current RCC therapeutics is well tolerated and effective in mice, and that future studies are warranted to determine the ability of acarbose to safely improve therapeutic responses in advanced renal cancer patients.

## 2. Results

### 2.1. Acarbose Blunts Postprandial Blood Glucose in Tumor-Free and Renal Tumor-Bearing Mice

As an inhibitor of oligosaccharide breakdown, acarbose slows the digestion of complex carbohydrates within the small intestine and lessens blood glucose spikes following a meal as depicted in Figure 1A. Therefore, patients managing hyperglycemia with acarbose take oral tablets at the start of each meal [27,28]. Given its unique mechanism of action, acarbose was ad-mixed into the diet in accordance with previous studies [23,24]. However, whereas prior studies used a NIH-31-based chow as the base diet (LabDiet 5LG6), we used a base diet (LabDiet 5LJ5) with a higher percentage of fat and lower percentage of carbohydrates to more closely resemble average macronutrient intake in United States adults according to NHANES reports [29]. Isocaloric 5LJ5 control (CON) and 5LJ5 acarbose-containing (ACA) diets contained the same macronutrient composition (Figure 1B). To verify the glucoregulatory properties of dietary acarbose, tumor-free BALB/c mice were fasted for four hours and then fed a controlled amount of either CON or ACA (Figure 1C). As expected, acarbose blunted blood glucose spikes after feeding (Figure 1C). We next interrogated the ability of ACA to control postprandial blood glucose spikes in mice bearing orthotopic renal tumors. As we have previously published, the left kidneys of BALB/c mice were injected with a syngeneic firefly luciferase-expressing renal cell carcinoma cell line (Renca-fLUC) [30]. Mice were randomized to CON or ACA following tumor challenge and fed ad libitum until day 15 when they were subjected to the same fast and re-feed assay as described in Figure 1C. ACA significantly blunted the rise in postprandial glucose in renal tumor-bearing mice relative to CON (Figure 1D), validating the previously described actions of acarbose [31], both in the absence and presence of renal tumor growth.

### 2.2. Acarbose Impedes the Growth of Murine Renal Tumors

We next evaluated the impact of ACA on renal tumor progression. Given the highly aggressive growth rate of orthotopic Renca tumors [30,32], mice were randomized to CON or ACA immediately following renal tumor challenge. Monitoring primary renal tumors over time by bioluminescent imaging (BLI) revealed that ACA impeded tumor outgrowth (Figure 2A). Of note, tumor BLI was similar between CON and ACA at day 6, illustrating that ACA did not impact renal tumor establishment (Appendix A). However, ACA consistently resulted in smaller excised tumor weights on days 21 and 25 following tumor challenge (Figure 2B). We also quantified spontaneous lung metastases by measuring BLI on excised lungs and observed no statistical difference in lung metastatic burden with ACA at either time point (Appendix A). These findings illustrated that although ACA impeded primary renal tumor growth, it did not protect against metastatic disease. Prior reports documented that tumor-free mice on ACA had lower body weights over time than mice on control diet, despite eating more food [23,24]. Therefore, we monitored murine body weights weekly during tumor growth. Orthotopic renal tumor outgrowth led to similar patterns of weight loss in mice on CON and ACA (Appendix A), demonstrating that ACA did not exacerbate weight loss. We also asked whether ACA impacted tumor growth when Renca tumor cells were injected subcutaneously, and found that ACA also impeded tumor growth when cells were injected in this alternate manner (Appendix A).

We next asked whether dietary composition influenced the ability of ACA to impair renal tumor growth by using an alternate NIH-31-based (5LG6) ACA diet (Appendix A), which was previously shown to extend longevity in mice relative to 5LG6 CON [24]. Renal tumor-bearing mice were subjected to either 5LG6 CON or 5LG6 + ACA, and then renal tumors were excised and weighed. Mice on 5LG6 + ACA exhibited lower renal tumor weights compared to 5LG6 CON (Appendix A), illustrating that acarbose impeded renal tumor growth across two different diet compositions.

### 2.3. Exogenous Glucose Reverses Acarbose-Induced Reductions in Renal Tumor Growth and Viability

We next sought to identify potential mechanisms driving ACA-induced reductions in renal tumor growth by addressing the impact of ACA on Renca tumor cell abundance, viability, and proliferation via flow cytometry at a day 21 time point. As previous studies demonstrated that Renca cells express the NKG2D ligand Rae-1γ [33], we classified renal tumor cells as CD45^−^/Rae-1γ^+^ (Figure 3A). As expected from our tumor weight data (Figure 2B), ACA significantly reduced the frequency of CD45^−^ Rae-1γ^+^ Renca cells in tumor-bearing kidneys (Figure 3A). We next used Ki67 expression (Appendix A) as a surrogate for tumor cell proliferation and observed a reduction in the frequency of proliferative tumor cells with ACA (Figure 3B). Further, analysis of tumor cell viability by dual Zombie Aqua and Annexin V staining revealed an increased proportion of late apoptotic (Zombie Aqua^+^/Annexin V^+^) Renca cells with ACA (63% vs. 43% in CON) (Figure 3C,D). Overall, these findings demonstrate that acarbose-induced reduction in primary renal tumor growth is associated with reduced tumor cell viability and proliferation.

To determine whether ACA-dependent reductions in systemically available glucose contributed to impaired tumor growth, we bypassed the primary gastrointestinal mechanism of action of acarbose by administering exogenous glucose directly to renal tumor-bearing mice on CON or ACA diets (Figure 3E). In the absence of exogenous glucose, ACA reduced renal tumor weights as shown previously; however, the addition of exogenous glucose led to tumor growth rebound in mice on ACA, resulting in excised tumor weights equivalent to those of mice on CON diet (Figure 3F). Glucose administration did not alter tumor growth in mice on CON (Figure 3F), suggesting sufficient glucose availability for robust tumor growth in these animals. Reductions in tumor weight with ACA and subsequent rescue with exogenous glucose corresponded with similar trends in the abundance of CD45^−^/Rae-1γ^+^ Renca tumor cells (Figure 3G). Importantly, ACA-associated increases in the frequency of dead/late apoptotic (Zombie Aqua^+^/Annexin V^+^) tumor cells were reversed in the presence of exogenous glucose (Figure 3H). Together, these findings illustrate that the addition of systemically available glucose overrides the anti-tumor activities of ACA, resulting in reduced tumor cell apoptosis and increased tumor growth.

### 2.4. The Anti-Tumor Activity of Acarbose Requires CD8 T Cells

As prior studies reported that glucoregulatory agents such as metformin exhibited anticancer effects in a CD8 T cell-dependent manner [17], we next asked whether the enhanced tumor control we observed with ACA depended on an intact CD8 T cell response. Here, mice were given orthotopic renal tumors, randomized to CON or ACA, and then subjected to CD8 T cell depletion by i.p. injection of anti-CD8β monoclonal antibodies (Figure 4A and Appendix A). In untreated mice, ACA again led to reduced tumor outgrowth by BLI versus CON. However, in the absence of CD8 T cells, mice on ACA and CON exhibited nearly identical patterns of renal tumor outgrowth, and CD8 T cell-depleted mice on ACA had accelerated tumor growth relative to CD8 T cell intact mice on ACA (Figure 4B). These observations were mirrored by renal tumor weights on day 21 (Figure 4C), demonstrating that the effect of ACA on renal tumor growth depended, at least in part, on an intact CD8 T cell response. We therefore quantified the frequency of CD8 TILs in mice on CON versus ACA on day 21 and observed a moderate but significant increase with ACA (Figure 4D). Further characterization revealed that CD8 TILs from mice on CON and ACA were phenotypically similar regarding their activation status (CD44^+^/CD62L^−^), PD-1 expression, and early-effector status (CD127^−^/KLRG1^−^) (Appendix A). Given that competition for glucose within the tumor microenvironment can impair CD8 TIL function [13], we next asked whether the glucoregulatory action of ACA compromised CD8 TIL effector function. In ACA groups, the production of effector molecules TNFα, IFNγ, perforin, and granzyme B remained intact versus CON (Figure 3E), illustrating no detectable detrimental effects of ACA on CD8 TIL functional capacity.

It was possible that ACA-associated elevations in CD8 TIL frequency on day 21 were confounded by the significant reduction in tumor weight with ACA at this time point. To address this possibility, we assessed CD8 TIL frequencies at two earlier time points (day 11 and day 15) when mice on CON and ACA exhibited similar tumor weights (Appendix A). While the frequency of CD8 TILs peaked on day 15 in both groups, mice on ACA exhibited elevated CD8 TIL influx on days 11, 15, and 21 (day 21 data repeated from Figure 4D for comparison over time) (Figure 4F). By day 25, CD8 TILs contracted similarly in both CON and ACA groups (Figure 4F). Considering our observation regarding increased tumor cell death with ACA, we next asked whether the earlier CD8 T cell influx corresponded to increases in tumor-associated antigen uptake by tumor-infiltrating dendritic cells (DCs). We therefore injected FITC-labeled latex microspheres into renal tumor-bearing kidneys on day 10 and assessed DC antigen uptake 24 h later using previously published methods (Figure 4G) [34,35]. Using the gating strategy in Appendix A, we identified an increased percentage of FITC^+^/CD11b^−^/CD11c^high^/I-A^d+^ DCs within tumor-bearing kidneys of mice on ACA (Figure 4H). We next employed a similar strategy to assess tumor-derived antigen uptake by injecting a replication-deficient adenoviral vector encoding green fluorescent protein (Ad5-GFP). Renca tumors are known to express the coxsackie adenovirus receptor (CAR) [36], which permits adenovirus-mediated expression of exogenous proteins. Intratumoral injection of Ad5-GFP resulted in tumor cell expression of GFP (Appendix A) and an increased frequency of GFP^+^ DCs within the tumor microenvironment with ACA (Figure 4I). Additionally, ACA led to increased expression of the T cell co-stimulatory ligand CD86 on intratumoral DCs, and elevated percentages of CD40^+^ DCs within renal tumor-draining lymph nodes on day 11 (Figure 4J). These findings correspond to increased CD8 TIL frequencies with ACA at the same time point. Collectively, we found that ACA-mediated reductions in renal tumor growth at least partially depend upon on heightened CD8 T cell anti-tumor immunity.

### 2.5. Combining Anti-PD-1 with Acarbose Impedes Lung Metastases

Based on our data thus far, we hypothesized that ACA would act as a therapeutic adjuvant for the ICI anti-PD-1, resulting in improved outcomes versus single-agent anti-PD-1. Mice were treated as in Figure 5A. Combined administration of ACA + anti-PD-1 yielded significantly reduced tumor weights relative to CON no therapy (NT) mice on day 21, whereas anti-PD-1 had a minimal effect on tumor burdens in mice on CON diet (Figure 5B). Although the average tumor weights from ACA + anti-PD-1 did not significantly differ from ACA NT or CON + anti-PD-1, treatment with ACA + anti-PD-1 resulted in 41% of mice achieving tumor weights within 0.2 g of an average tumor-free kidney (as designated by the dotted line) (Figure 5B,C). In comparison, mice from CON NT, CON + anti-PD-1, and ACA NT had 0%, 0%, and 16% of tumors within this range, respectively (Figure 5C). Quantification of CD8 TIL functional capacity from CON NT, CON + anti-PD-1, ACA NT, and ACA + anti-PD-1 revealed respective stepwise increases in the frequencies of IFNγ^+^ (Appendix A) TNFα^+^, and perforin^+^ CD8 T cells within the tumor microenvironment (Figure 5D). Overall, these findings support the idea that ACA administered in combination with anti-PD-1 promotes anti-tumor immunity in a manner sufficient to impede primary tumor growth in a subset of mice, using a renal cancer model that is resistant to anti-PD-1 monotherapy.

As anti-PD-1 immunotherapy is used to treat metastatic kidney cancer, we also evaluated the effect of ACA + anti-PD-1 on lung metastases by BLI assessment on day 21 (Figure 5E). Strikingly, combination treatment led to statistically significant decreases in lung metastases versus all other groups, and a 17-fold reduction in lung metastatic burden versus NT CON (Figure 5E). The combination of ACA + anti-PD-1 also led to elevated frequencies of TNFα^+^ and perforin^+^ CD8 T cells within metastatic lungs (Figure 5F), suggesting an improvement in systemic anti-tumor immunity. Overall, ACA led to improved outcomes following anti-PD-1 treatment with respect to both primary tumor weights and metastatic tumor burdens in the lungs.

### 2.6. Acarbose Improves Response to Rapamycin

To determine whether the improvements observed with ACA + anti-PD-1 extended to targeted therapies approved for use in metastatic RCC, we combined ACA with the mTOR inhibitor rapamycin or vehicle control (VC) administered in the drinking water (Figure 6A). Treatment with rapamycin led to significant reductions in renal tumor weights in mice on either CON or ACA compared to their respective VC groups (Figure 6B), resulting in similar percentages (65% and 76%, respectively) of tumors within 0.2 g of a tumor-free kidney (Appendix A). This effect did not require CD8 T cells, as their depletion did not alter rapamycin-induced reduction in tumor weights (Figure 6B), even though rapamycin led to increased frequencies of CD8 TILs in both CON and ACA groups (Appendix A) and an improved capacity for IFNγ production by CD8 TILs with ACA+ rapamycin (Appendix A). Interestingly, treatment with rapamycin in mice on CON did not alter lung metastases as quantified by BLI (Figure 6C). However, ACA + rapamycin led to a significant reduction in lung metastases versus CON VC, ACA VC, and CON + rapamycin (Figure 6C). Although reduction in primary tumor growth did not require CD8 T cell involvement (Figure 6B), depletion of CD8 T cells largely reversed the ACA + rapamycin-mediated reductions in lung metastases and led to a rebound in lung metastatic burden (Figure 6C). Moreover, ACA+ rapamycin induced the greatest expansion in the frequency of TNFα^+^ and perforin^+^ (Figure 6D) CD8 T cells within metastatic lungs. These results demonstrate that ACA can improve outcomes following targeted mTOR inhibitor administration in a mouse model of orthotopic renal cancer.

## 3. Discussion

Despite the multiple successes of ICIs and targeted therapeutics in RCC, tumors from many patients do not respond to these treatments. Ongoing research strives to improve responses by identifying and targeting mechanisms of therapeutic resistance. Evidence in favor of using dietary interventions or antidiabetic drugs to improve therapeutic responses is compelling yet limited, as this remains an emerging field. A full understanding of the impact of these agents on tumor growth and therapeutic responses is becoming crucial, as they offer tolerable, inexpensive, and easily accessible options for combination therapy. Many studies document the anticancer activities of the antidiabetic drug metformin; however, these effects are reported to occur through a variety of glucose-independent mechanisms [17,19,20,21,37]. We therefore investigated the potential anti-tumor activities of the glucoregulatory drug acarbose, an alternative antidiabetic agent with a distinct mode of action and caloric restriction mimetic (CRM) properties.

Here, we administered dietary acarbose to tumor-bearing mice in the absence and presence of therapy and compared various outcomes to mice fed an isocaloric macronutrient-matched control diet. In our orthotopic renal cancer model, we observed that acarbose improved therapeutic responses to both anti- PD-1 checkpoint blockade and the mTOR inhibitor rapamycin. Specifically, the combination of ACA + anti-PD-1 led to nearly complete primary tumor growth inhibition in 41% of mice versus 0% of mice on control diet. Importantly, combining either anti-PD-1 or rapamycin with acarbose generated the greatest reduction in spontaneous lung metastases. This finding coincides with recent reports that used alternative dietary/nutrition-based interventions in tumor-bearing mice, which observed improved responses to chemotherapy alone [5] or chemotherapy + ICI [26], upon combination with CRMs or 48 h fasting. Another similar study observed a combinatorial additive effect of short-term starvation or a fasting-mimicking diet and chemotherapy in a murine mammary carcinoma model [38]. Metformin, another antidiabetic agent, synergized with anti-PD-1 in murine models of melanoma and colon cancer [19] and everolimus in breast cancer cells and mammary xenografts [39]. However, both of these studies used only subcutaneous tumor models and did not examine the impact of their combinatorial treatments on metastatic events. Our study concurs with and expands upon these findings by using an orthotopic renal tumor model and quantifying spontaneous lung metastases in response to combinatorial administration of acarbose with either anti-PD-1 or rapamycin. The antimetastatic properties of anti-PD-1 or rapamycin in combination with acarbose is of particular interest, given the high mortality rates of metastatic RCC versus localized disease. Therefore, we identify acarbose as a novel therapeutic adjuvant for further investigation in metastatic preclinical renal cancer.

In addition to improving therapeutic efficacy, acarbose alone impeded primary renal tumor growth; however, the impact of nutrition-based interventions or glucoregulatory agents on tumor growth in the absence of therapy remains unclear in other models. Although our finding in the Renca renal cancer model recapitulates several studies that found reduced tumor growth in mice on metformin alone across multiple subcutaneous tumor models [17,37], other studies did not observe this [19]. Similarly, CRMs alone do not significantly impede tumor growth in murine fibrosarcoma [5], but short-term starvation (48 h) or feeding mice a fasting-mimicking diet slowed tumor growth in triple-negative murine mammary carcinoma [38]. We therefore posit that the effects of these interventions on tumor growth in the absence of therapy depend on tumor type, anatomic location, and the intervention itself.

Acarbose-induced inhibition of renal tumor growth depended in part on systemic reductions in glucose availability. Human renal tumors may possess a particular vulnerability to glucose limitation, as RCC displays highly glycolytic metabolism [15], and human renal tumors with the highest expression of glycolysis-related genes were characterized by low CD8 T cell infiltration [16]. Elevated tumor cell glycolysis has also been identified as a mechanism of resistance to T cell-based immunotherapy in other tumor models, including melanoma [10,11]. This illustrates the potential of glucose limitation as an avenue for combinatorial therapeutic intervention in RCC.

Acarbose-induced reductions in tumor growth also depended on CD8 T cells, as depletion of these cells led to tumor growth rebound. We further observed increased frequencies of CD8 T cells within tumors of mice on acarbose, which reflects similar observations in tumor-bearing mice on metformin [17,19]. Improvements in anti-tumor immunity in response to nutrition-based interventions have also been reported by other groups. For example, the combination of short-term fasting or CRMs with chemotherapy agents leads to CD8 T cell-dependent reductions in tumor growth and depletion of regulatory T cells within tumors [5], and increased expression of inducible T cell co-stimulator on tumor-infiltrating CD8 T cells in mice, indicative of a more activated phenotype [26]. Administration of a fasting-mimicking diet in combination with chemotherapy also promotes CD8 T cell infiltration into murine mammary tumors to impede tumor growth [38]. Together, these findings illustrate a link between nutrition-based interventions and enhanced anti-tumor immunity. Although our study and others found no evidence of detrimental effects on CD8 T cells, it is important to note a potential consequence of nutrient limitation on CD8 T cell function, as glucose deprivation [13] or blocking T cell glycolysis [40] can impair cytokine production and function in CD8 T cells. With acarbose, we observed increased CD8 TILs at multiple time points, increased antigen uptake by intratumoral DCs, and increased expression of DC activation markers. When considered with our observation that acarbose promoted increased tumor cell death, we posit a scenario in which glucose limitation by acarbose leads to increased tumor cell death and release of tumor-associated antigens for recognition by antigen presenting cells, which drives increased antigen uptake/presentation, and leads to subsequent CD8 T accumulation within renal tumors. Enhanced CD8 T cell anti-tumor immunity with acarbose impedes renal tumor growth, serving as a therapeutic adjuvant for anti-PD-1 and rapamycin.

Although our study provides an initial analysis of the effects of acarbose on tumor growth, anti-tumor immunity, and response to immune-based and targeted therapies, it also contains limitations. For example, our tumor model does not include specific antigens or antigen-specific T cells, and our experiments were conducted only with female mice. Further, our study included an evaluation of acarbose specifically in renal cancer. The use of nondiabetic, lean mice illustrates the potential for use of acarbose in normoglycemic cancer patients but does not address the potential contributions of acarbose in hyperglycemic individuals, a frequent comorbidity known to increase risk for RCC in women [41] and worsen survival outcomes for RCC patients [42]. Acarbose use in these patients may be particularly beneficial. Additionally, we found that glucose regulation, tumor cell death, and anti-tumor immunity cooperated to reduce tumor growth and improve therapeutic responses with acarbose; however, this does not preclude additional mechanisms for the effects of acarbose in this setting. Given these limitations and the fact that our study is the first to examine acarbose in this context, we acknowledge that our study is hypothesis testing as well as hypothesis generating, and more studies are needed.

Our study supports the idea that limiting glucose availability with acarbose in renal tumors may confer improved susceptibility to immune-based and targeted therapies. This raises the question of whether diets that mimic this effect, such as low-carbohydrate diets or high-fiber diets, could also improve responses. Future studies should address the impact of glucoregulatory or dietary interventions as potential therapeutic adjuvants in multiple tumor models to advance guidance regarding the nutritional advice that is presented to cancer patients.

Another area for future investigation involves the effects of acarbose on the gut microbiome. Several recent findings demonstrate the importance of the gut microbiome on ICI responses, where enrichment of “good” bacteria correlated with favorable responses to ICIs [43,44,45]. In addition to reducing postprandial glucose excursions, acarbose lengthens the exposure of carbohydrates to commensal bacteria in the gut, altering the gut microbiome in mice and humans [46,47,48]. Of note, type II diabetes patients on acarbose had an enrichment of gut *Bifidobacterium longum* [49], a population recently associated with improved ICI responses in melanoma patients [45]. Although our study did not examine the gut microbiome, future studies with acarbose and alternate dietary interventions should address this important question to identify agents capable of favorably remodeling the gut microbiome.

## 4. Materials and Methods

### 4.1. Animals and Diets

Wild-type BALB/c female mice were purchased from Charles River (NCI-Frederick colony) at 7–8 weeks of age. Mice were randomly separated into cages at a density of 4–5 per cage, allowed to acclimate for one week following receipt, and were fed an NIH-31 open-formula source diet during this period. Research involving mice was approved by the Institutional Animal Care and Use Committee (IACUC) at the University of Alabama at Birmingham on 1 March 2018 (IACUC-21147). Mice were housed 4–5 per cage in ventilated cages in a specific pathogen-free status room and were subjected to a 6:00 a.m.–6:00 p.m. 12 h light cycle followed by a 12 h dark cycle. Except where indicated, mice were fed ad libitum. Control diet (LabDiet^®^ 5LJ5) or 5LG6 control diet (LabDiet^®^ 5LG6) was purchased through Purina LabDiet^®^. Acarbose was purchased from Tecoland Corporation (Irvine, CA, USA). Acarbose-containing diet was prepared by TestDiet^®^ at a ratio of 1 g of acarbose per kilogram of base LabDiet^®^ 5LJ5 or 5LG6 diet (0.1% acarbose). Acarbose-containing diet was dyed dark blue by TestDiet^®^ to differentiate from control diet. Both control and acarbose-containing diets were irradiated. Where indicated, mice were randomized to CON or ACA immediately following tumor challenge (on day 0).

### 4.2. Fast and Re-Feeding Assay and Exogenous Glucose Administration

Tumor-free BALB/c mice were maintained on NIH-31 diet for one week. Mice were fasted at the beginning of their dark cycle from 6:00 p.m. to 10:00 p.m. (4 h). Fasting blood glucose levels were measured from the tail vein using a glucometer and test strips (ReliOn). Mice were randomized and fed 1 g of either CON or ACA. Subsequent blood glucose readings were measured every 15 min for 1 h. If mice did not consume any food during this time, they were excluded from analysis (*n* = 1/10 excluded for tumor-free mice, 3/10 excluded for tumor-bearing mice). Renal tumor-bearing mice were on CON or ACA for 14 days prior to a 4 h fast and re-feed assay. For exogenous glucose experiments, mice on CON and ACA were given 3 intraperitoneal injections of D-glucose at a dose of 2 g/kg on days 10, 13, and 16 following tumor challenge.

### 4.3. In Vivo Tumor Modeling

The Renca renal carcinoma cell line (derived from and syngeneic to BALB/c mice) was purchased from ATCC, engineered to express firefly-luciferase through lentiviral transduction, and cultured as previously described [32,34,50,51]. Cells were not authenticated after purchase. Cells were confirmed negative for mycoplasma, passaged, and used at the same passage number (passage 13) in all tumor studies described here to limit experimental variation. Intrarenal tumor challenges were performed where indicated as previously described [32,50,51]. BALB/c mice were orthotopically injected with 5 × 10^4^ live Renca cells in a volume of 100 μL/mouse. Tumor burdens were assessed from live anesthetized mice or from excised lungs by bioluminescent imaging (BLI) using intraperitoneal injections with 1 mg of luciferin and an IVIS Lumina III imager from Perkin Elmer, provided by the Small Animal Imaging Facility at UAB. In the rare event that BLI yielded negative radiance values when imaging lungs (*n* = 3 lungs), the negative value was replaced with the background BLI radiance value for tumor-free lungs (1.5 × 10^4^ photons/s/cm^2^/sr). Primary renal tumors were excised and weighed.

### 4.4. Flow Cytometry

Spleens, renal tumors, and renal tumor-draining lymph nodes were analyzed by flow cytometry to assess relevant immune populations. Spleens and tumors were homogenized in 5 mL 1X HBSS using the spleen cycle on the gentleMACS™ Dissociator from Miltenyi Biotec. Tumor-draining lymph nodes were homogenized manually with frosted-glass slides. When assessing only T cell populations, tumor and spleen homogenates were passed through a 70 μm filter to yield single-cell suspensions. When staining for dendritic cells, homogenized spleen and tumor tissues were enzymatically digested with 5 μg/mL Liberase and 37.5 μg/mL of DNase I at 37 °C for 15 min (spleens) or 30 min (tumors) with gentle rotation in a shaking incubator prior to filtration. Cells were counted and resuspended in 1X HBSS in accordance with cell counts and stained in a 96-well round-bottom plate (approximately 4 × 10^6^ cells per well). Cells were then stained with fixable Zombie Aqua viability dye, Fc-blocking antibody (αCD16/32, Biolegend, San Diego, CA, USA), and fluorophore-conjugated antibodies. For quantification of intracellular proteins, the Cytofix/Cytoperm kit from BD was used. To assess T cell cytokine/effector molecule production, whole-tumor single-cell suspensions at a cell density of 1 × 10^5^ cells per well were stimulated with anti-CD3 and anti-CD28 (8 μg/mL and 10 μg/mL, respectively) in a 96-well flat-bottom plate. Cells were incubated for 4 h and GolgiPlug was added during the last two hours of stimulation, and then cells were harvested and stained in accordance with the protocols listed above for surface and intracellular markers. Annexin V staining was performed in accordance with the BD Annexin V apoptosis detection kit protocol (BD Biosciences, Franklin Lakes, NJ, USA).

### 4.5. In Vivo Antibodies

For CD8 depletion studies in BALB/c mice on CON and ACA, anti-mouse CD8β antibody (BioXcell, clone 53-5.8, Lebanon, NH, USA) was injected intraperitoneally (i.p.) at a dose of 100 μg/mouse on days 1, 3, 5, 7, 14, and 20 following tumor challenge. For CD8 depletion groups in rapamycin studies, anti-mouse CD8β antibody was given i.p. at a dose of 100 μg/mouse on days 4, 6, 7, 14, and 20 following tumor challenge as previously published [32]. For anti-PD-1 studies, anti-mouse PD-1 antibody (BioXcell, clone RMP1-14) was administered i.p. at a dose of 250 μg/mouse on days 10, 13, and 16 following tumor challenge.

### 4.6. Rapamycin Studies

Rapamycin was purchased from LC Laboratories (Woburn, MA, USA) and dissolved in ethanol (100%) at a concentration of 15 mg/mL. Beginning on day 7 post-tumor challenge, stock rapamycin was diluted 1:1000 in drinking water, and drinking water was changed weekly. Dosing rapamycin in this manner yields an approximate dose of 1.5 mg/kg/day [52]. Mice on vehicle control received ethanol in drinking water at a 1:1000 dilution.

### 4.7. Intratumoral Injections

FITC-labeled microspheres (Fluoresbrite^®^ YG, 1.0 µm) were purchased from Polysciences (Warrington, PA, USA). Replication-deficient adenoviral vector encoding GFP was purchased from the University of Iowa Viral Vector Core (Iowa City, IA, USA) Where indicated, 100 μL of undiluted microspheres or 10^9^ pfu of Ad5-GFP diluted in sterile saline were injected into growing orthotopic renal tumors on day 10 post-tumor challenge and mice were sacrificed 24 h later.

### 4.8. Statistical Analysis

All statistical analyses were performed using Prism 8 (GraphPad Software, Version 8.4,0 (671), San Diego, CA, USA). Gaussian distribution of data was assessed using Shapiro–Wilk normality testing. For studies examining repeated-measures over time between 2 or more groups, two-way repeated-measures analysis of variance (ANOVA) was performed, followed by post-hoc comparison-*t*-tests with Bonferroni correction. For studies involving 2 independent groups, *t*-tests or nonparametric Mann–Whitney U tests were performed as appropriate. For 3 or more independent groups, one-way ANOVA with Tukey’s post-hoc test or nonparametric Kruskal–Wallis tests with Dunn’s post-hoc test were performed as appropriate. All tests were two tailed. Throughout, asterisks designated significance (* *p* < 0.05; ** *p* < 0.01; *** *p* < 0.001; **** *p* < 0.0001); ns = not significant. Nonsignificant trending *p*-values of *p* ≤ 0.15 are indicated; otherwise, “ns” is used.

## 5. Conclusions

To conclude, we provide the first report, to our knowledge, that the antidiabetic agent acarbose can impede renal tumor growth, promote anti-tumor immunity, and improve outcomes to anti-PD-1 and mTOR inhibitor-based therapeutics in preclinical renal cancer. Reductions in tumor growth depended on the glucoregulatory action of acarbose, as well as CD8 T cell involvement. Our study expands upon prior findings that nutrition-based interventions can improve responses to multiple treatment modalities and adds acarbose to the list of potential immunotherapeutic adjuvants. Our data suggest that combining acarbose with anti-PD-1 or targeted mTOR inhibitors improves therapeutic responses in mice with renal cancer. Further studies are needed, including retrospective and prospective clinical studies, to determine the translational relevance of our findings in patients with metastatic RCC.

## Figures and Tables

**Figure 1 cancers-12-02872-f001:**
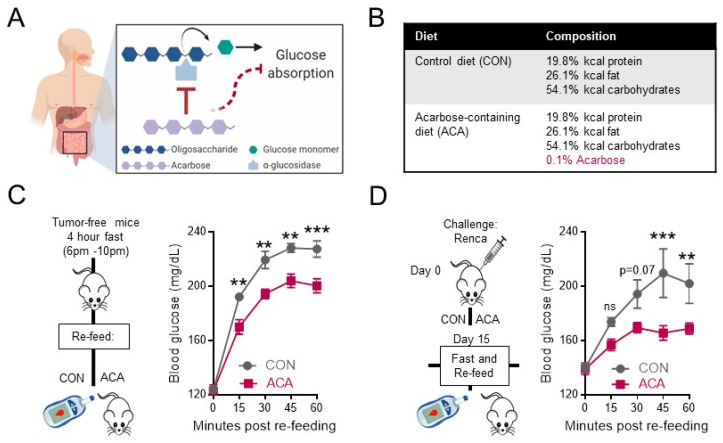
Acarbose blunts postprandial blood glucose levels in the absence and presence of renal tumor growth. (**A**) Graphical representation of the mechanism of action for acarbose. In the presence of acarbose, the breakdown of oligosaccharides into glucose is delayed due to competitive inhibition of alpha-glucosidases within the small intestine. (**B**) Macronutrient composition percentages of kilocalories (kcal) of CON and ACA diets. (**C**) Experimental design for fast and re-feed assay (left) and blood glucose measurements from tail vein of tumor-free BALB/c mice (right, *n* = 5 mice per group, 1 representative experiment). (**D**) Experimental design for orthotopic renal tumor challenge (left) and day 15 post-tumor challenge blood glucose measurements from tail vein of BALB/c mice (right, CON: *n* = 3 mice; ACA: *n* = 7 mice, 2 experiments). Data are average curves and are presented as the means ±SEM. Statistical differences in panels C and D were calculated using two-way repeated-measures ANOVA and post-hoc *t*-test with Bonferroni correction for multiple comparisons (ns = not significant; ** *p* < 0.01; *** *p* < 0.001; trending *p*-values of *p* < 0.15 are shown; otherwise, “ns” is shown). CON = control diet; ACA = acarbose-containing diet.

**Figure 2 cancers-12-02872-f002:**
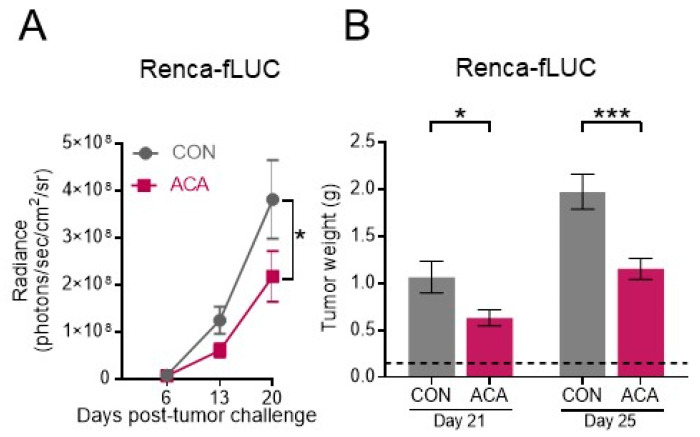
Acarbose impedes growth of renal tumors. (**A**) Primary renal tumor burdens over time measured by bioluminescent imaging (BLI, CON: *n* = 13 mice; ACA: *n* = 16 mice, 3 experiments). (**B**) Excised renal tumor weights on day 21 (CON: *n* = 16 mice; ACA: *n* = 20 mice, 4 experiments) and 25 (CON: *n* = 19 mice; ACA: *n* = 23 mice, 4 experiments) post-tumor challenge. Dotted line represents the average weight of tumor-free kidneys. Data in (**A**) and (**B**) are group averages and are presented as the means ±SEM. Statistical differences in panel (**A**) were calculated using two-way repeated-measures ANOVAs and post-hoc *t*-test with Bonferroni correction for multiple comparisons. Statistical differences in panel **B** were determined by independent *t*-tests or nonparametric Mann–Whitney tests as appropriate (ns = not significant; trending *p*-values of *p* < 0.15 are shown; otherwise, “ns” is shown. * *p* < 0.05; *** *p* < 0.001). CON = control diet; ACA = acarbose-containing diet.

**Figure 3 cancers-12-02872-f003:**
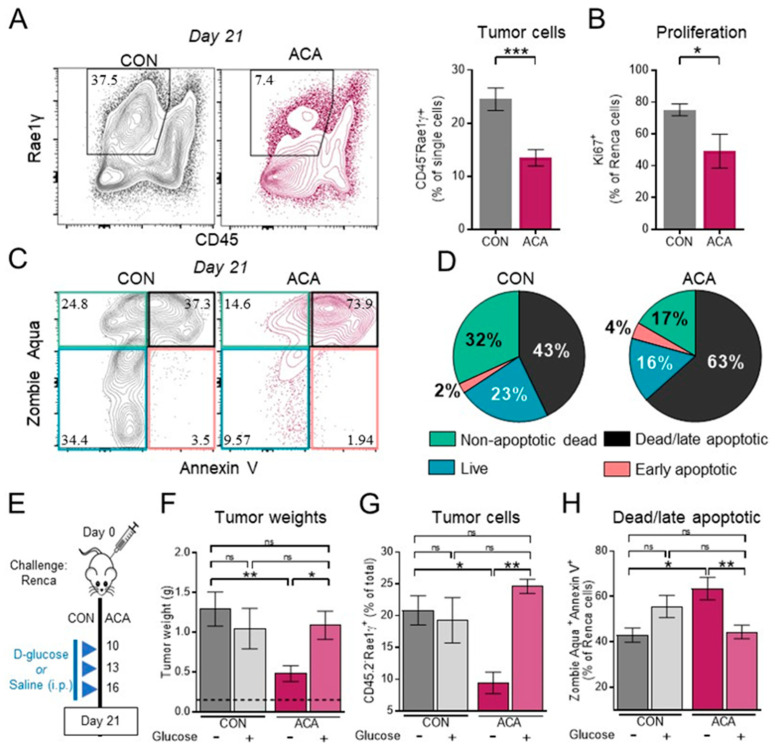
Exogenous glucose restores renal tumor growth and tumor cell viability in mice on acarbose. (**A**) (left) Representative flow cytometry plots of Renca tumor cells from day 21 whole renal tumors (Ancestry: FSC-A vs. SSC-A > single cells > live) and (right, CON: *n* = 8 mice; ACA: *n* = 10 mice, 2 experiments) frequency of CD45^−^Rae1γ^+^ renal tumor cells from day 21 whole renal tumors. (**B**) Ki67^+^Renca tumor cells from day 21 tumors (CON: *n* = 13 mice; ACA: *n* = 9 mice, 2 experiments). Ancestry: FSC-A vs. SSC-A > single cells > live > CD45^−^Rae1γ^+^. (**C**) Representative flow cytometry density plots for Zombie Aqua and Annexin V staining (Ancestry: FSC-A vs. SSC-A > single cells> CD45^−^Rae1γ^+^). (**D**) Pie charts showing proportions of dead/late apoptotic (Zombie Aqua^+^/Annexin V^+^), nonapoptotic (Zombie Aqua^+^/Annexin V^−^), early apoptotic (Zombie Aqua^−^/Annexin V^+^), and live (Zombie Aqua^−^/Annexin V^−^) Renca tumor cells for CON (*n* = 3 mice, 1 representative experiment) and ACA (*n* = 5 mice, 1 representative experiment). (**E**) Experimental strategy for exogenous glucose administration. Mice were intrarenally (i.r.) tumor challenged and randomized to CON +/− glucose or ACA +/− glucose. Mice were injected i.p. with D-glucose on days 10, 13, and 16. (**F**) Day 21 excised renal tumor weights from mice on CON, ACA, CON + glucose or ACA + glucose (*n* = 10, 10, 14, and 10 mice, respectively; 2–3 experiments). Dotted black line represents the average weight of tumor-free kidneys. (**G**) Day 21 frequency of renal tumor cells within whole renal tumors (CON: *n* = 3 mice, ACA: *n* = 5 mice; 1 representative experiment). Ancestry: FSC-A vs. SSC-A > single cells > live. (**H**) Day 21 frequency of dead/late apoptotic Renca tumor cells (CON: *n* = 3 mice, ACA: *n* = 5 mice; 1 representative experiment). Ancestry: FSC-A vs. SSC-A > single cells> CD45^−^Rae1γ^+^. Data are summary data, and where applicable, data are presented as the means ±SEM. Statistical differences in panels (**A**) and (**B**) were determined by *t*-tests or nonparametric Mann–Whitney tests as appropriate. Statistical differences in exogenous glucose experiments (**F**–**H**) were determined using one-way ANOVA with Tukey’s post-hoc test or nonparametric Kruskal–Wallis test with Dunn’s post-hoc test as appropriate (ns = not significant; trending *p*-values of *p* < 0.15 are shown; otherwise, “ns” is shown. * *p* < 0.05; ** *p* < 0.01; *** *p* < 0.001). CON = control diet; ACA = acarbose-containing diet.

**Figure 4 cancers-12-02872-f004:**
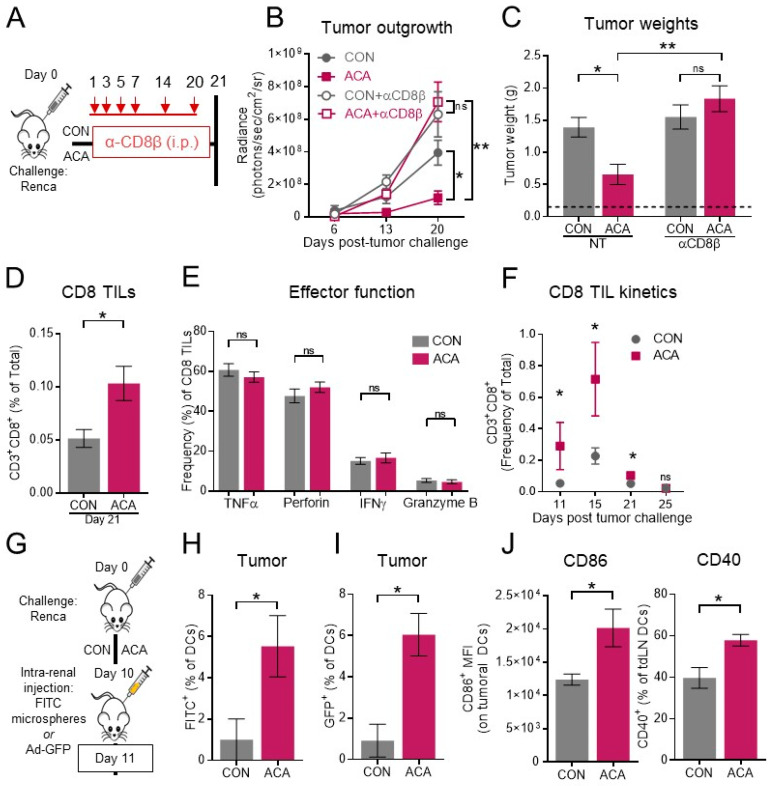
Acarbose-mediated reductions in renal tumor weight require CD8 T cell involvement. (**A**) Experimental strategy for CD8 depletion studies. Mice were i.r. tumor challenged and randomized to CON +/− CD8 depletion or ACA +/− CD8 depletion. Mice were injected i.p. with α-CD8β on days 1, 3, 5, 7, 14, and 20 post-tumor challenge. (**B**) Primary renal tumor outgrowth over time as measured by BLI (CON: *n* = 5 mice, ACA: *n* = 5 mice, CON + αCD8β: *n* = 10 mice, ACA+ αCD8β, *n* = 10 mice; 1 representative experiment). (**C**) Day 21 excised renal tumor weights (CON: *n* = 9 mice, ACA: *n* = 10 mice, CON + αCD8β: *n* = 7 mice, ACA+ αCD8β, *n* = 9 mice; 2 experiments). Dotted black line represents the average weight of tumor-free kidneys. (**D**) Frequency of CD8 T cells within day 21 renal tumors (CON: *n* = 23 mice, ACA: *n* = 30 mice; 4 experiments). Ancestry: FSC-A vs. SSC-A > single cells > live, CD11b^−^CD11c^−^ dump gate. (**E**) Intratumoral CD8 T cell production of TNFα, perforin, IFNγ, and granzyme B from ex vivo restimulated whole-tumor homogenates (CON: *n* = 16 mice, ACA: *n* = 18 mice; 2 experiments). (**F**) Frequency of intratumoral CD8 T cells on days 11 (CON: *n* = 5 mice, ACA: *n* = 5 mice; 1 experiment), 15 (CON: *n* = 8 mice, ACA: *n* = 7 mice; 2 experiments), 21 (CON: *n* = 23 mice, ACA: *n* = 30 mice; 4 experiments), and 25 (CON: *n* = 14 mice, ACA: *n* = 13 mice; 2 experiments). Postrenal tumor challenge. Day 21 data are repeated from panel D for comparison over time. (**G**) Experimental strategy for FITC-microsphere or Ad5-GFP injection studies, where i.r. tumor-bearing mice on CON or ACA were i.r. injected on day 10 with either FITC-microspheres or Ad5-GFP. (**H**) Frequency of intratumoral DCs expressing FITC on day 11 (*n* = 5 mice per group, 1 experiment). (**I**) Frequency of GFP^+^ DCs within day 11 renal tumors (*n* = 4 mice per group, 1 experiment). (**J**) Left: CD86 mean fluorescence intensity on day 11 intratumoral DCs (*n* = 4 mice per group, 1 experiment). Ancestry: FSC-A vs. SSC-A > single cells > live > CD45^+^ >CD11b^−^ > I-A^d+^CD11c^high^ > CD86^+^. Right: Frequency of CD40^+^ DCs from renal tumor-draining lymph nodes. Ancestry: FSC-A vs. SSC-A > single cells > live > CD45 ^+^ > CD11b^−^ > I-A^d+^CD11c^high^. Data are summary data and are presented as the means ±SEM. Statistical differences in panel (**B**) were calculated using two-way mixed-effects ANOVA and Dunnett’s post-hoc test for multiple comparisons using CON groups as reference. Statistical differences in **C** were determined using nonparametric Kruskal–Wallis test with Dunn’s post-hoc test. Statistical differences in panels (**C**–**J**) were determined using *t*-tests or nonparametric Mann–Whitney tests as appropriate (ns = not significant; trending *p*-values of *p* < 0.15 are shown; otherwise, “ns” is shown. * *p* < 0.05; ** *p* < 0.01). CON = control diet; ACA = acarbose-containing diet; NT = no therapy; Ad-GFP = adenoviral vector encoding GFP; tdLN = tumor-draining lymph node DC = dendritic cell.

**Figure 5 cancers-12-02872-f005:**
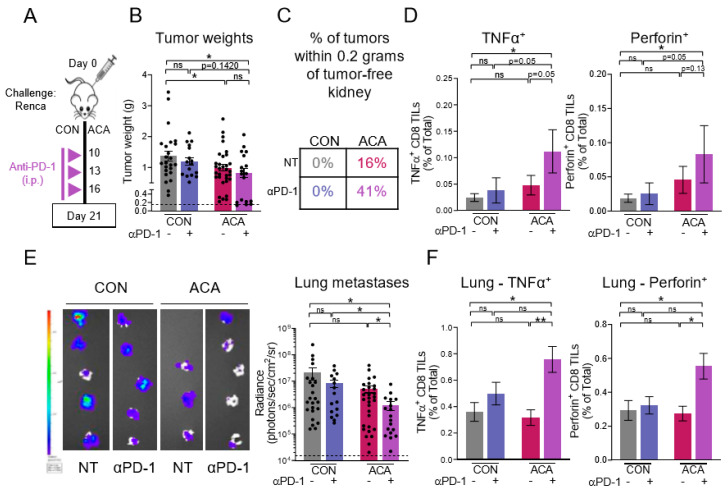
Combination of acarbose and anti-PD-1 improves therapeutic response. (**A**) Experimental design for anti-PD-1 experiments. Mice were i.r. tumor challenged and randomized to CON +/− anti-PD-1 or ACA +/− anti-PD-1. Mice were injected i.p. with anti-PD-1 on days 10, 13, and 16 post-tumor challenge. (**B**) Day 21 excised renal tumor weights (CON, NT: *n* = 24 mice, CON + anti-PD-1: *n* = 16 mice, ACA, NT: *n* = 31 mice, ACA, anti-PD-1: *n* = 18 mice; 2–4 experiments). Dotted black line represents the average weight of tumor-free kidneys. (**C**) Percentage of tumors shown in (**B**) within 0.2 g of a tumor-free kidney from each groups. (**D**) Frequencies of TNFα^+^ and perforin^+^ CD8 T cells within day 21 renal tumors, respectively (CON, NT: *n* = 10 mice, CON + anti-PD-1: *n* = 7 mice, ACA, NT: *n* = 11 mice, ACA, anti-PD-1: *n* = 7 mice; 2 experiments). (**E**) Representative BLI images from day 21 excised lungs (left) and quantification of metastatic burden measured by BLI within day 21 excised lungs on a logarithmic scale (right; CON, NT: *n* = 24 mice, CON + anti-PD-1: *n* = 16 mice, ACA, NT: *n* = 31 mice, ACA, anti-PD-1: *n* = 18 mice; 2–4 experiments). Dotted black line represents the average BLI of tumor-free lungs. (**F**) Frequencies of TNFα^+^ and perforin^+^ CD8 T cells within day 21 metastatic lungs, respectively (CON, NT: *n* = 10 mice, CON + anti-PD-1: *n* = 7 mice, ACA, NT: *n* = 9 mice, ACA, anti-PD-1: *n* = 8 mice; 2 experiments). Data from individual mice are shown in (**B**) and (**E**). Data are presented as the means ±SEM. Statistical differences were determined using nonparametric Kruskal–Wallis test with Dunn’s post-hoc test (ns = not significant; trending *p*-values of *p* < 0.15 are shown; otherwise, “ns” is shown. * *p* < 0.05; ** *p* < 0.01;). CON = control diet; ACA = acarbose-containing diet; NT = no therapy.

**Figure 6 cancers-12-02872-f006:**
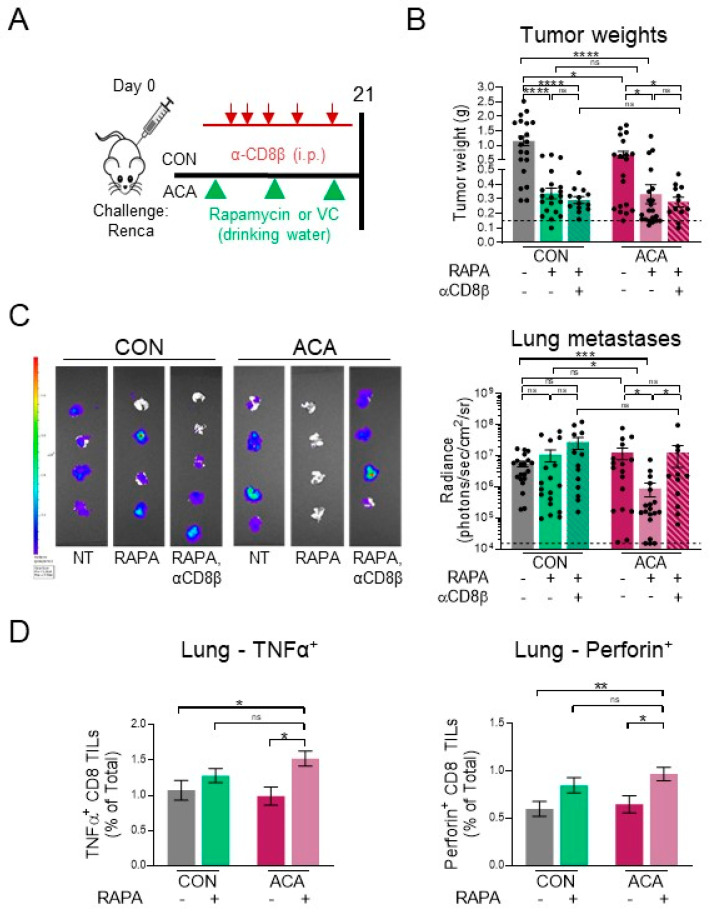
Efficacy of rapamycin is improved with acarbose. (**A**) Experimental design for rapamycin experiments. Mice were i.r. tumor challenged and randomized to CON +/− rapamycin, ACA +/− rapamycin, CON + rapamycin + CD8 depletion, and ACA + rapamycin + CD8 depletion. Rapamycin or vehicle control (VC) were administered in drinking water beginning on day 7 post-tumor challenge. Drinking water was changed weekly. A subset of mice was injected i.p. with α-CD8β on days 4, 6, 7, 14, and 20 post-tumor challenge. (**B**) Day 21 excised renal tumor weights (CON VC: *n* = 20 mice, CON + rapamycin: *n* = 20 mice, CON + rapamycin + αCD8β: *n* = 13 mice, ACA VC: *n* = 19 mice, ACA + rapamycin: *n* = 21 mice, ACA + rapamycin+ αCD8β: *n* = 13 mice; 2–4 experiments). Dotted black line represents the average weight of tumor-free kidneys. (**C**) Representative BLI images from day 21 excised lungs (left). Quantification of metastatic burden measured by BLI within day 21 excised lungs on a logarithmic scale (right, CON VC: *n* = 20 mice, CON + rapamycin: *n* = 18 mice, CON + rapamycin + αCD8β: *n* = 13 mice, ACA VC: *n* = 17 mice, ACA + rapamycin: *n* = 19 mice, ACA + rapamycin+ αCD8β: *n* = 11 mice; 2-4 experiments). Dotted black line represents the average BLI of tumor-free lungs. (**D**) Frequencies of TNFα^+^ and perforin^+^ CD8 T cells within day 21 metastatic lungs, respectively (CON VC: *n* = 9 mice, CON + rapamycin: *n* = 8 mice, ACA VC: *n* = 8 mice, ACA + rapamycin: *n* = 9 mice; 2 experiments). Data from individual mice are shown in (**B**) and (**C**). Summary data are shown in (**D**). Data are presented as the means ±SEM. Statistical differences were determined using one-way ANOVA with Tukey’s post-hoc test or nonparametric Kruskal–Wallis test with Dunn’s post-hoc test (ns = not significant; trending *p*-values of *p* < 0.15 are shown; otherwise, “ns” is shown. * *p* < 0.05; ** *p* < 0.01; *** *p* < 0.001; **** *p* < 0.0001). CON = control diet; ACA = acarbose-containing diet; NT = no therapy; RAPA = rapamycin; VC = vehicle control.

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
