# Peer review of "The Antidiabetic Agent Acarbose Improves Anti-PD-1 and Rapamycin Efficacy in Preclinical Renal Cancer"

_cancers, 2020, doi:10.3390/cancers12102872_

Round 1

Reviewer 1 Report

The study is carefully designed and performed accordingly.
The explanation of each step hypothesis vs. study design vs. results is very clear and easy to read even for a non-expert reader.
Graphs, images and flow charts are clear, easy to read and to understand.
Treatment of metastatic renal cell carcinoma made a dramatic change in the recent years. From death within few months to a prolonged survival time measured in years due to "biological" medications. Indeed despite the dramatic change some patients will benefit for several months before progression, hence understanding the mechanism of action and augmenting the effect is crucial to our arsenal of treatments.

Author Response

We thank the reviewer for their thoughtful comments.

Reviewer 2 Report

The antidiabetic agent acarbose improves anti-PD-1 and rapamycin efficacy in pre-clinical renal cancer by Orlandella et al.

Strengths:

  • Well written study
  • In this manuscript, the authors explored a novel therapeutic approach for advanced renal cell carcinoma that warrant further investigation. They performed extensive experimental analysis and demonstrated that combining acarbose with the current renal cell carcinoma therapeutics (either anti-PD-1 or rapamycin), can reduce lung metastases and improve therapeutic efficacy. This interesting study can be the basis for the development of new therapeutic modalities in renal cell carcinoma that can positively impact the RCC patients’ management and improve the therapeutic outcome.

Weaknesses:

  1. Line 111 Figure 1 (A) Graphical representation of the mechanism of action for acarbose, please describe the figure briefly.
  2. Line 116 Figure 1 legend (Data are average curves for n=3-7 mice per group from 1-2 independent experiments), please specify how many mice per each group as well as how many experiments for each. Same for line 150, 151, 199, 200, 266, 311, 312
  3. Line 194Figure 3 (E) Experimental strategy for exogenous glucose administration, please describe the experiment briefly. Same for line 254 Figure 4 (A), line 304 Figure 5 (A), line 336 Figure 6 (A)
  4. Few grammatical and typo errors that can be easily fixed.

Author Response

The antidiabetic agent acarbose improves anti-PD-1 and rapamycin efficacy in pre-clinical renal cancer by Orlandella et al.

Strengths:

  • Well written study
  • In this manuscript, the authors explored a novel therapeutic approach for advanced renal cell carcinoma that warrant further investigation. They performed extensive experimental analysis and demonstrated that combining acarbose with the current renal cell carcinoma therapeutics (either anti-PD-1 or rapamycin), can reduce lung metastases and improve therapeutic efficacy. This interesting study can be the basis for the development of new therapeutic modalities in renal cell carcinoma that can positively impact the RCC patients’ management and improve the therapeutic outcome.

We thank the reviewer for these comments.

Weaknesses:

  1. Line 111 Figure 1 (A) Graphical representation of the mechanism of action for acarbose, please describe the figure briefly.
  2. Line 116 Figure 1 legend (Data are average curves for n=3-7 mice per group from 1-2 independent experiments), please specify how many mice per each group as well as how many experiments for each. Same for line 150, 151, 199, 200, 266, 311, 312
  3. Line 194Figure 3 (E) Experimental strategy for exogenous glucose administration, please describe the experiment briefly. Same for line 254 Figure 4 (A), line 304 Figure 5 (A), line 336 Figure 6 (A)
  4. Few grammatical and typo errors that can be easily fixed.

We thank the reviewer for these comments. We have addressed each of these comments to improve clarity regarding the methods and experimental design.

Action taken: We have made the requested changes to each of the following points, detailed below. **Note that line numbers for revisions were denoted when “all markup” is shown on the word document; if viewing as “simple markup”, line numbers will be slightly different.

  1. Line 111 Figure 1 (A) Graphical representation of the mechanism of action for acarbose, please describe the figure briefly. The figure legend now contains a brief description for the mechanism of action for acarbose (lines 111-113)
  2. Line 116 Figure 1 legend (Data are average curves for n=3-7 mice per group from 1-2 independent experiments), please specify how many mice per each group as well as how many experiments for each. Same for line 150, 151, 199, 200, 266, 311, 312. All figure legends now contain more specific details as to how many mice were used per group and how many experiments were used to build each figure, and the previous details have been deleted for clarity. See changes on the following lines: 116, 118, 151-153, 192, 194, 200, 204, 206, 208, 268-272, 274-277, 281-283, 327, 331, 334, 337, 368-370, 373-375, and 376-377.
  3. Line 194Figure 3 (E) Experimental strategy for exogenous glucose administration, please describe the experiment briefly. Same for line 254 Figure 4 (A), line 304 Figure 5 (A), line 336 Figure 6 (A). Figure legends for figure 3-6 now contain experiment descriptions where indicated to improve clarity of experimental design. See changes on the following lines: 201-203, 265-267, 279-280, 325-327, and 364-368.
  4. Few grammatical and typo errors that can be easily fixed. We have revised the document for these errors and made changes accordingly (see tracked changed on revised document).

Reviewer 3 Report

In the manuscript, Rachael et al, investigated the antitumor effect of Acarbose in a pre-clinical renal cancer model. Major findings of the study show that Acarbose induced reduction in primary renal tumor growth, which is associated with reduced tumor cell viability and proliferation, though the outcome was reversed by exogenous glucose administration. Further, the author showed that the delayed tumor outgrowth in Acarbose treated mice occurred in a CD8 dependent manner. Interestingly, combined therapy of Acarbose with either anti-PD-1 or rapamycin significantly reduced lung metastases. Overall, this study is well designed and provides a new approach for better clinical outcomes on cancer treatments. The data presentation and the results are clear and convincing. The text is well written and has relevant references.

Minor comments.

  • Flow figures- the author may indicate the percentage of each gated populations of the flow dot plots for better reading.
  • Line 454, why did the author choose only female mice for the study.

Author Response

In the manuscript, Rachael et al, investigated the antitumor effect of Acarbose in a pre-clinical renal cancer model. Major findings of the study show that Acarbose induced reduction in primary renal tumor growth, which is associated with reduced tumor cell viability and proliferation, though the outcome was reversed by exogenous glucose administration. Further, the author showed that the delayed tumor outgrowth in Acarbose treated mice occurred in a CD8 dependent manner. Interestingly, combined therapy of Acarbose with either anti-PD-1 or rapamycin significantly reduced lung metastases. Overall, this study is well designed and provides a new approach for better clinical outcomes on cancer treatments. The data presentation and the results are clear and convincing. The text is well written and has relevant references.

We thank the reviewer for these comments. Responses to minor comments indicated below.

Minor comments.

  • Flow figures- the author may indicate the percentage of each gated populations of the flow dot plots for better reading.

Action Taken: We have added the percentages to the gated populations in the flow plots in Figure 3A and Figure 3C.

  • Line 454, why did the author choose only female mice for the study.

This is an excellent question. Our lab also studies diet-induced obesity in BALB/c mice, which involves placing mice on high fat diet for 20 weeks to induce obesity as BALB/c mice are more resistant to diet-induced obesity. In these studies, we use female mice because male mice fight and bite each other when caged together for so long, which is not ideal when studying immune responses. Because of this, our model of orthotopic kidney cancer is optimized in female mice, and we therefore began this acarbose study using female mice. However, there are reports of sex differences with acarbose, so we conducted an exploratory study in male mice. Unfortunately, this study was conducted in March and we were unable to perform repeat experiments due to the pandemic and limited resources at the time, so we cannot draw conclusions from this single experiment. This would be an interesting follow up study.

Action Taken: In the discussion, we have added this as a limitation of our study (458-459)***Note that line numbers for revisions were denoted when “all markup” is shown on the word document; if viewing as “simple markup”, line numbers will be slightly different.)